# Timing to Intubation COVID-19 Patients: Can We Put It Off until Tomorrow?

**DOI:** 10.3390/healthcare10020206

**Published:** 2022-01-21

**Authors:** Júlio César Garcia de Alencar, Juliana Martes Sternlicht, Alicia Dudy Muller Veiga, Julio Flávio Meirelles Marchini, Juliana Carvalho Ferreira, Carlos Roberto Ribeiro de Carvalho, Izabel Marcilio, Katia Regina da Silva, Vilson Cobello Junior, Marcelo Consorti Felix, Luz Marina Gomez Gomez, Heraldo Possolo de Souza, Denis Deratani Mauá

**Affiliations:** 1Emergency Department, Hospital das Clínicas HCFMUSP, Faculdade de Medicina, Universidade de São Paulo, São Paulo 05403-000, Brazil; juliana.sternlicht@fm.usp.br (J.M.S.); alicia.muller@fm.usp.br (A.D.M.V.); julio.marchini@hc.fm.usp.br (J.F.M.M.); luz.marina@hc.fm.usp.br (L.M.G.G.); heraldo.possolo@fm.usp.br (H.P.d.S.); dr.julioalencar@gmail.com (E.U.C.G.); 2Divisão de Pneumologia, Instituto Do Coração, Hospital das Clínicas HCFMUSP, Faculdade de Medicina, Universidade de São Paulo, São Paulo 05403-000, Brazil; juliana.ferreira@hc.fm.usp.br (J.C.F.); carlos.carvalho@hc.fm.usp.br (C.R.R.d.C.); 3The Center for Emergencies in Public Health, Salvador 40301-155, Brazil; izabel.marcilio@hc.fm.usp.br; 4Instituto Do Coração, Hospital das Clínicas HCFMUSP, Faculdade de Medicina, Universidade de São Paulo, São Paulo 05403-000, Brazil; katia.regina@hc.fm.usp.br; 5Corporative IT, Hospital das Clínicas, Faculdade de Medicina, Universidade de São Paulo, São Paulo 05403-000, Brazil; vilson.cobello@hc.fm.usp.br; 6Data Office, Hospital das Clínicas, Faculdade de Medicina, Universidade de São Paulo, São Paulo 05403-000, Brazil; marcelo.consorti@hc.fm.usp.br; 7Institute of Mathematics and Statistics, Universidade de São Paulo, São Paulo 05403-000, Brazil; denis.maua@usp.br; 8Hospital das Clínicas HCFMUSP, Faculdade de Medicina, Universidade de São Paulo, São Paulo 05403-000, Brazil; julioalencar.cdl@gmail.com

**Keywords:** SARS-CoV-2, COVID-19, intubation, emergency medicine

## Abstract

Background: The decision to intubate COVID-19 patients receiving non-invasive respiratory support is challenging, requiring a fine balance between early intubation and risks of invasive mechanical ventilation versus the adverse effects of delaying intubation. This present study analyzes the association between intubation day and mortality in COVID-19 patients. Methods: We performed a unicentric retrospective cohort study considering all COVID-19 patients consecutively admitted between March 2020 and August 2020 requiring invasive mechanical ventilation. The primary outcome was all-cause mortality within 28 days after intubation, and a Cox model was used to evaluate the effect of time from onset of symptoms to intubation in mortality. Results: A total of 592 (20%) patients of 3020 admitted with COVID-19 were intubated during study period, and 310 patients who were intubated deceased 28 days after intubation. Each additional day between the onset of symptoms and intubation was significantly associated with higher in-hospital death (adjusted hazard ratio, 1.018; 95% CI, 1.005–1.03). Conclusion: Among patients infected with SARS-CoV-2 who were intubated and mechanically ventilated, delaying intubation in the course of symptoms may be associated with higher mortality. Trial registration: The study protocol was approved by the local Ethics Committee (opinion number 3.990.817; CAAE: 30417520.0.0000.0068).

## 1. Introduction

Patients with severe COVID-19 present with dyspnea and hypoxemia [1]. A portion of these patients may develop Acute Respiratory Distress Syndrome (ARDS), which is defined as the acute onset of bilateral infiltrates, severe hypoxemia, and lung edema that is not fully explained by cardiac failure or fluid overload [2]. Therapeutic strategies for these patients require respiratory support, ranging from supplementary oxygen through nasal cannula, non-rebreather mask, high-flow nasal oxygen cannula (HFNO), and non-invasive ventilation (NIV) to endotracheal intubation (ETI), invasive mechanical ventilation (MV), and extracorporeal membrane oxygenation (ECMO) [3,4,5].

Several guidelines have advocated early ETI and MV in the management of COVID-19-related ARDS. These guidelines stated that ETI and MV must be established as soon as non-invasive respiratory interventions fail to maintain O_2_ saturation (SpO_2_) above 90% or to reduce inspiratory efforts [6,7,8]. Delaying intubation in patients with ARDS leads to worse outcomes when compared to intubation at the time of admission to Intensive Care Unit (ICU) or diagnosis of ARDS [9,10]. Moreover, postponed intubation may increase the risk of peri-procedural cardiac arrest, whether delaying results in severe hypoxemia and lack of preoxygenation [11,12,13].

Regardless of the definition of early or late intubation (i.e., based on a specific time threshold from the onset of symptoms, ICU admission, or of a prior trial of HFNO/NIV), clinicians caring for patients with COVID-19 seem to become eager to favor a wait-and-see strategy over time, tolerating use of non-invasive oxygen support and lower SpO_2_ for more extended periods [14].

In spite of minimize complications is a constant goal in the care of these patients, mechanical ventilation hold several risks, such endotracheal-tube complications, volume-induced alveolar injury, barotrauma, ventilation-associated pneumonia and sepsis, dysfunction and atrophy of respiratory muscles, and hemodynamic complications related to positive pressure and sedation [15,16,17,18].

Therefore, physicians must carefully consider the decision to submit these patients to ETI and MV. Considering that there is no consensus about the best moment to indicate endotracheal intubation and mechanical ventilation for severe COVID-19 patients in respiratory failure, we performed a retrospective analysis of these patients to determine the association between time of intubation during the disease evolution and outcomes.

## 2. Materials and Methods

### 2.1. The Aim, Design, and Setting of the Study

We conducted a retrospective unicentric cohort study from March to August 2020 at Emergency Department (ED) in the Hospital das Clínicas da Universidade de São Paulo (HC-FMUSP), a 2200-bed academic center with 900-bed unit dedicated to COVID-19 patients.

We included all consecutive adult patients (≥18 years) with confirmed COVID-19 and mechanical invasive ventilation requirements. Confirmed COVID-19 was defined as at least one positive result on reverse transcriptase-polymerase chain reaction (RT-PCR) obtained from nasopharyngeal swabs or bronchial secretions.

We excluded patients already intubated at admission, patients with Hospital-Acquired SARS-CoV-2 infection, defined as date of hospitalization before the onset of symptoms, and patients with missing dates.

There were no established protocols or predefined criteria for intubation in our institution. The decision to intubate has been left to the discretion of the Emergency Physicians or Intensivists who responded in accordance with patients’ individual needs and clinical status. Most of the clinicians practiced early intubation strategy during the beginning of the pandemic. However, as more data emerged and recommendations changed, clinicians were more comfortable monitoring patients on non-invasive modes of oxygenation (such as HFNO or NIV). Nowadays, the clinician’s judgment to initiate mechanical ventilation was influenced by a multitude of factors, including oxygen saturation, respiratory rate, work of breathing, mental status, and hemodynamics.

The study protocol was approved by the local Ethics Committee (opinion number 3.990.817; CAAE: 30417520.0.0000.0068), which also waived the need for written informed consent. We adhere to STROBE guidelines.

### 2.2. Study Protocol and Outcomes

Patient data were collected through electronic medical records, and a database was built on REDCap^®^ software (v11.0.3, Vanderbilt University, Nashville, TN 37235, United States). The variables studied were sex, age, COVID-19 RT-PCR results, date of admission, date of symptoms, SpO_2_ at admission, date of intubation, and outcomes (death or discharge).

The primary outcome was all-cause mortality within 28 days after intubation.

### 2.3. Statistical Analysis

Continuous data were described as means and standard deviations (SDs) when the distribution is normal and medians and interquartile ranges (IQRs) for non-normally distributed data. Categorical variables are described as the number of events and proportions.

*T*-tests were used to compare variables with normal distribution, Wilcoxon rank-sum tests were used for comparisons of non-normally distributed continuous variables, and Chi-square tests were used for comparisons of categorical variables.

For the primary outcome analysis, we present a Kaplan–Meier survival plot for the univariate analysis, along with an unadjusted Cox model to examine the relationship between the day of intubation (counting from the onset of symptoms) and 28-day mortality after the ETI. We then fit a multivariable Cox model adjusting for age, sex, the number of comorbidities, respiratory rate at admission, peripheral oxygen saturation on room air at admission, serum urea at admission, and C-reactive protein levels at admission and the time (in days) after the COVID symptoms first appeared until the day of the intubation. These variables make up the 4C score [19]. We had demonstrated that the 4C score was an accurate predictor of outcomes in our population [20]. In this present study, we did not have access to the Glasgow coma scale, and we did not use this variable for adjusted analyses.

We evaluated the proportional hazards assumption with Schöenfeld residuals and log-log plots. Effect estimates from the Cox models are presented as hazard ratios.

Moreover, we also performed a multivariable logistic regression model to examine the relationship between individual baseline characteristics (using 4C variables at admission) and outcomes. Multivariable models were then created to investigate the association between the day of intubation and mortality 28-days after the procedure after adjusting for differences in patients at admission.

Two expert mathematicians (LMGG and DDM) performed this data analysis. For all analyses, the null hypothesis was evaluated at a two-sided significance level of 0.05, with the calculation of 95% confidence intervals.

Statistical analyses were performed using StataCorp^®^ (Release 13, StataCorp LP, College Station, TX, USA) 2013 and using R version 4.0.3 (R Core Team, Vienna, Austria).

## 3. Results

### 3.1. Patients

Our hospital was designated to be the reference for severe COVID-19 cases during the pandemic. A total of 3020 consecutive COVID-19 patients were admitted during the study period, all of them classified as critically ill, according to WHO guidelines [4]. For our analysis, we evaluated all 592 patients with Severe Acute Respiratory Syndrome caused by SARS-CoV-2 virus who were submitted to EIT and MV in our institution (Figure 1).

All these 592 patients were treated in the ICU, under the institution protocol. Mortality after 28 days was high in this group (52.4%) and non-survivors were older and had more comorbidities than survivors, as observed in Table 1. Other severity markers were low lymphocytes, low platelet count, and impaired renal function at admission.

### 3.2. Timing of Intubation and Mortality

The median time from onset of symptoms to intubation was 9 days (IQR 7–14). For each additional day between the onset of symptoms and intubation, the unadjusted hazard ratio for death within the follow-up period was 1.006 (95% CI, 1.006–1.030) (Figure 2).

We performed a multivariable Cox model adjusting in 530 patients, because 62 (11.6%) observations were excluded again due to missing data. Among these, 272 patients died. After adjustment for all covariates, the hazard ratio was 1.018 (95% CI, 1.018–1.03) (Figure 3).

According to the COX analysis, multivariate logistic regression adjusted by 4C score at admission demonstrated a statistically significant relationship between the day of intubation and mortality with an increase of OR 1.04 per day. The effect of time to intubation on 28-day mortality was 0.0076 [95% CI 0.001–0.014], which indicates an average increase of 0.8% in the probability of death per day.

Lastly, we performed a post-hoc exploratory analysis using COX proportional hazard regression model adjustment for 4C variables and SAPS 3 at admission, and the hazard ratio was 1.015 (95% CI, 1.01 1.025) (Table 2).

## 4. Discussion

Choosing the exact moment to submit a patient with COVID-19 in respiratory distress to endotracheal intubation and mechanical ventilation is still a challenge to emergency physicians and intensivists. In this cohort study, we demonstrate that the time of intubation affects the outcomes, and the adjusted hazard ratio for mortality was 1.018 for each day of delay in intubation following the onset of symptoms.

These results are in accordance with reports about timing of intubation in patients with ARDS of other etiologies. In ICU patients that met ARDS criteria, patients submitted to mechanical ventilation later presented higher 60-day mortality compared to patients who were intubated at the moment of ARDS diagnosis [10]. Interestingly, in this study, patients in early intubation group had higher APACHE II when compared with the late-intubation group, suggesting that delaying mechanical ventilation was responsible for unfavorable outcomes [10].

In a very influential editorial, published during the first months of the pandemic, early ETI and MV was advocated for COVID-19 ARDS [21]. The authors reasoned that the spontaneous vigorous inspiratory effort produced by these patients could lead to increased high transpulmonary pressures and consequent lung damage. This hypothesis was challenged soon after [22], and the issue is still unsolved. A recent meta-analysis evaluated 12 studies and could not find evidence that timing of intubation have any effect in outcomes [14]; however, the included studies were heterogeneous and a definitive conclusion could not be reached. Moreover, all the studies enrolled patients at ICUs and some of them had very few patients in each arm.

Fayed et al., in a single-center retrospective study with 110 patients, concluded that the timing of intubation for patients with severe COVID-19 pneumonia was not significantly associated with overall mortality. However, post hoc analysis of patients who had SOFA scores between 0 and 9 showed that significantly more patients within this score range who were intubated late died compared to those who were intubated early. Additionally, Kaplan–Meier survival analysis stratified by SOFA score also showed that patients who were intubated later were more likely to die [23].

Our data present a clear distinction between patients who were submitted to ETI and MV as soon as the respiratory failure were made and the ones who were submitted to the procedures later. The Kaplan–Meier curves show an increase in death probability each day the ETI is delayed. This finding was maintained even when all the corrections were performed to exclude confounding factors.

There are some possible explanations for these findings. First, and most important, the data presented here are from the first year of the pandemic. It was not clear the best moment to perform ETI in COVID-19 patients, and our institution did not have a policy about it at that time. Second, we need to point out the fear of serious complications during ETI. Although we have shown the safety of the procedure in COVID-19 [24], there was still the idea that keeping the patients out of mechanical ventilation would be safer. Additionally, there was a high track record of intubated patients in our country; COVID-19 patients requiring MV have had an unacceptable mortality in Brazil [25] and, although our hospital was better prepared for the pandemic than the rest of the country, the idea that patients in mechanical ventilation had a higher risk of death could have entered doctors’ minds. Finally, it is possible that patients who were submitted to delayed intubation were the ones more severely ill, and the procedure was postponed due to the patients’ worsening condition. For example, ETI could be delayed because the patient was hemodynamic unstable and would not tolerate sedation and positive pressure. This is less probable, since the ICU time after intubation was similar between the two groups.

This work has some limitations, and these data must be interpreted with caution. First, given the retrospective nature of this study, we are unable to determine whether delayed intubation itself resulted in the observed increase in mortality. It is possible that the procedure postponement resulted in fewer intubated patients overall, leaving only the most severely ill patients in the cohort receiving mechanical ventilation. We reduced this bias by performing an analysis adjusted by the 4C score on admission. Second, we could not determine the factors that influenced the clinicians’ decision regarding the timing of intubation with this study design. Finally, the retrospective data used were not sufficiently granular to examine the impact of pharmacologic therapies prior to versus after intubation.

On the other hand, this study has some very important strengths. The number of patients enrolled is higher than most of the studies approaching this issue. The fact that it was performed in a single center is more a strength than a weakness. Our hospital was dedicated to assist only severe COVID-19 patients. At the peak of the pandemic, we had 350 ICU beds for these patients. The track record of our ICUs were comparable to other centers around the world [26], confirming that the ICU care was not determinant for the outcome.

Besides those factors, the adjusted analyzes using 4C at admission were performed by two mathematicians using the COX model and logistic regression. Data demonstrated the association between the day of intubation and mortality risk. Finally, contrary to other works about the issue, we counted the timing of intubation based on the start of the symptoms and not on the day of admission, which brings more weight to our findings.

## 5. Conclusions

This combination of findings provides some support for the conceptual premise that increasing the time from onset of symptoms to intubation in severe COVID-19 patients is associated with higher mortality. Some of the issues emerging from this finding relate specifically to supportive care consisting of early intubation, which may be associated with better outcomes.


## Figures and Tables

**Figure 1 healthcare-10-00206-f001:**
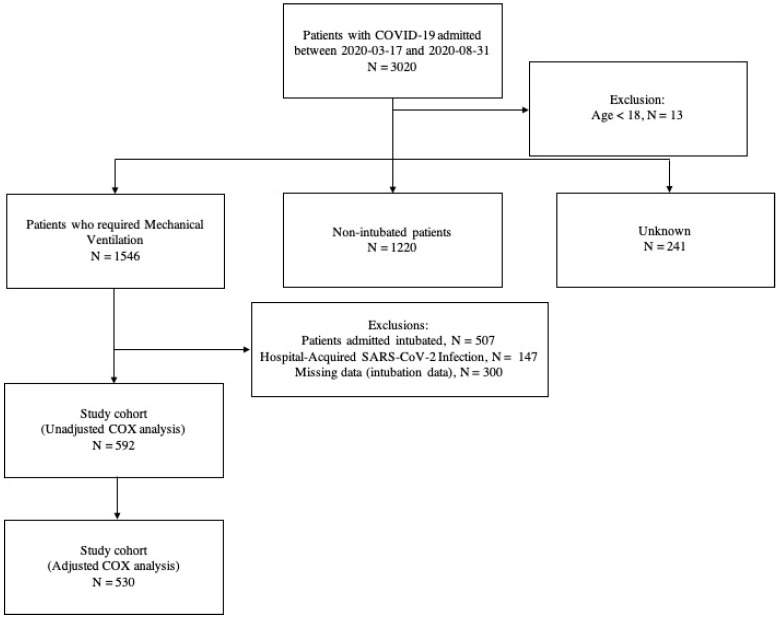
Study cohort.

**Figure 2 healthcare-10-00206-f002:**
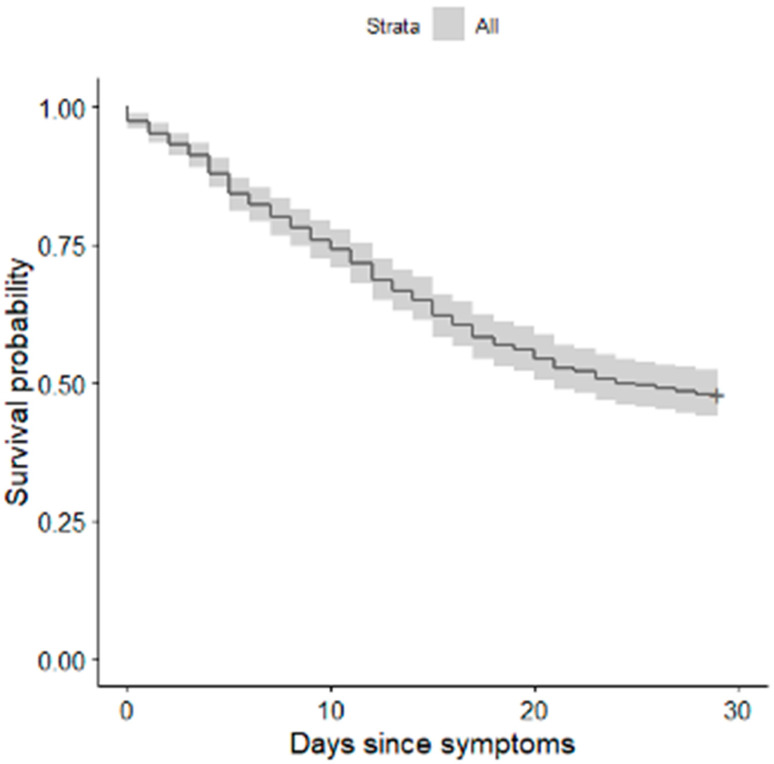
Kaplan–Meier survival plot for 28-day mortality in relation to intubation time.

**Figure 3 healthcare-10-00206-f003:**
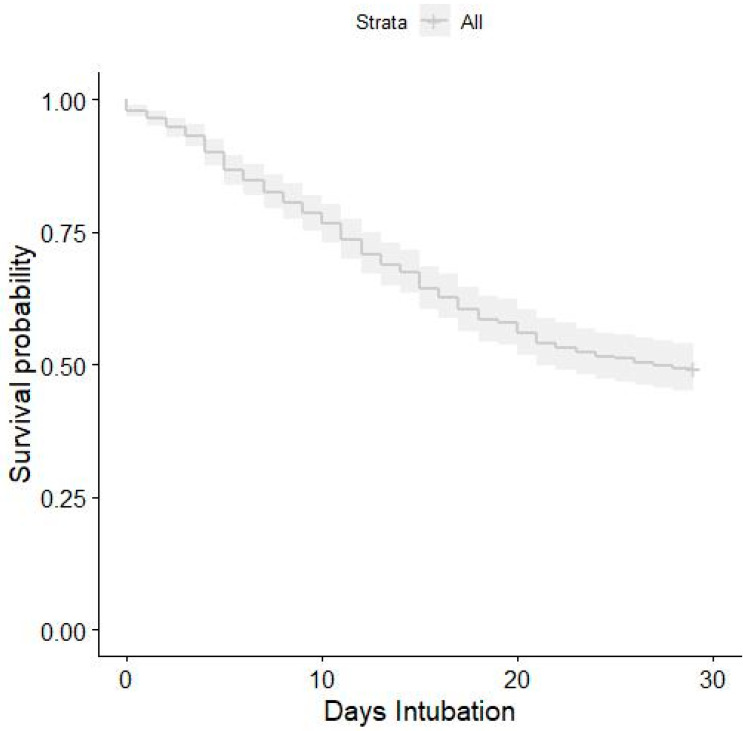
Kaplan–Meier survival plot for 28-day mortality in relation to intubation time after the 4C score at admission adjustments.

**Table 1 healthcare-10-00206-t001:** Baseline characteristics according to outcomes.

Sample Characteristic	Survivors	Non-Survivors	*p* Value *
Sample Size (*n*)	282	310	-
Age, year	56.8 (44.9–65.6)	64.8 (55.8–72.3)	<0.001
Sex (Male)	132 (22.3)	214 (36.1)	<0.001
Status at Admission			
Respiratory Rate, Breaths/min	26 (22–32)	25 (20–30)	0.09
Oxygen Saturation, %	93 (90–95)	93 (89–96)	0.63
Heart Rate, Beats/min	90 (80–101)	92 (80.3–101)	0.4
SAPS 3	61 (50–70.5)	67 (54.5–77)	<0.001
Blood Tests			
Leucocytes, Thousand Per mm^3^	9.0 (6.9–12.4)	8.8 (5.9–12.5)	0.27
Lymphocytes, thousand per mm^3^	0.9 (0.7–1.2)	0.7 (0.5–1.1)	<0.001
Platelets, Thousand Per mm^3^	222 (173.8–294.3)	195 (140.5–262)	<0.001
Creatinine, mg/dL	0.9 (0.7–1.4)	1.1 (0.8–2.0)	<0.001
Urea, mg/dL	38 (25–59)	53 (35–84)	<0.001
C-reactive Protein, mg/dL	182.7 (104.1–281.8)	168.8 (94.1–256.7)	0.24
D-dimer, μg/mL	1531 (876.8–4551)	1692 (964–5485)	0.39
Lactate Dehydrogenase, U/L	483 (359.8–611.5)	519 (390.3–673.3)	0.04
Comorbidities, *n* (%)			
Hypertension	139 (23.5)	226 (38.2)	<0.001
Diabetes	92 (15.5)	148 (25.0)	<0.001
Dialysis	59 (10.0)	174 (29.4)	<0.001
Cancer	8 (1.4)	43 (7.3)	<0.001
Immunodeficiency	7 (3.0)	15 (6.3)	<0.001
Current Smoker	22 (3.7)	40 (6.8)	<0.001
Obesity	89 (15.3)	81 (14.0)	<0.001

**Table 2 healthcare-10-00206-t002:** Hazard ratio multivariable Cox mode.

	Hazard Ratio	*p*-Value	95% CI for the Hazard Ratio	
Sex (male)	1.235	0.111	0.953	1.601
Age, year	1.018	0.001	1.007	1.028
C-Reactive Protein, mg/dL	0.998	0.010	0.997	1.000
Respiratory Rate, Breaths/min	0.983	0.076	0.965	1.002
Oxygen Saturation, %	0.992	0.339	0.976	1.008
Urea, mg/dL	1.005	0.000	1.003	1.008
Days since Symptoms	1.022	0.000	1.010	1.034
SAPS 3	1.015	0.002	1.005	1.025

## Data Availability

The data that support the findings of this study are available from the corresponding author (J.C.G.d.A.) upon reasonable request.

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
