# Peer review of "Timing to Intubation COVID-19 Patients: Can We Put It Off until Tomorrow?"

_healthcare, 2022, doi:10.3390/healthcare10020206_

Round 1

Reviewer 1 Report

Dear Authors,

Can I kindly ask you to address my concerns? All details you can easily find below.

  • page 2, line 50-54: vv ECMO is the broadly accepted therapeutic tool in ARDS. Please address this in the manuscript with relevant reference
  • page 2, line 65-67: you should properly present complications of invasive mechanical ventilation namely barotrauma, volutrauma, atelectrauma, biotrauma etc. This paragraph should be properly rebuild
  • page 2, line 74-75: please address the potential complications related to prolonged HFNOT
  • page 2, line 86-91: it is not necessary to describe detailed structure of your institution, this paragraph is too long. '2200-bed academic center with 900-bed unit dedicated to covid-19 patients' could be enough
  • page 4, line 176-188: Again this section is too long. Do not duplicate detailed results here. This paragraph should be much more condensed. Just present briefly significant observations.
  • page 5, line 237-238: As I realize your study concerns not typical ED or ICU but rather covid-19 unit. Please delete this sentence as it is not necessary and informative at all.
  • page 5, line 242-244: Standardization of therapy does not mean that every patient receives the same medication or is mechanically ventilated in the exactly the same way (parameters). Usually the therapeutic strategy is individualized to some extent. Please change the meaning of this paragraph as it is incorrect.

Author Response

Dear Revisor,

Thank you for your suggestions, comments and ideas for manuscript improvements!
Below we provide a detailed point-by-point response to you. Original comments will be provided and our response will be immediately below.

1. Page 2, line 50-54: vv ECMO is the broadly accepted therapeutic tool in ARDS. Please address this in the manuscript with relevant reference

Thank you for point this out. We added the study: "Extracorporeal membrane oxygenation for severe acute respiratory distress syndrome associated with COVID-19: a retrospective cohort study"

2. Page 2, line 65-67: you should properly present complications of invasive mechanical ventilation namely barotrauma, volutrauma, atelectrauma, biotrauma etc. This paragraph should be properly rebuild

Thank you for point this out. We have rewritten this paragraph and added a professor Tobin reference.
Correct text: In spite of minimize complications is a constant goal in the care of these patients, mechanical ventilation hold several risks, such endotracheal-tube complications, vol-ume-induced alveolar injury, barotrauma, ventilation-associated pneumonia and sepsis, dysfunction and atrophy of respiratory muscles, and hemodynamic complications related to positive pressure and sedation (14–16) (17).  

3. Page 2, line 74-75: please address the potential complications related to prolonged HFNOT.

Thank you for point this out! We have rewitten the Introduction and we excluded this sentence. While we believed HFNOT complications would be an excellent discussion, we had afraid to stray from the topic.

4. Page 2, line 86-91: it is not necessary to describe detailed structure of your institution, this paragraph is too long. '2200-bed academic center with 900-bed unit dedicated to covid-19 patients' could be enough

Thank you for point this out. We rewrote the paragraph.
Corrected text:  We conducted a retrospective unicentric cohort study from March to August 2020 at Emergency Department (ED) in Hospital das Clínicas da Universidade de São Paulo (HC-FMUSP), a 2200-bed academic center with 900-bed unit dedicated to COVID-19 pa-tients.

5. Page 4, line 176-188: Again this section is too long. Do not duplicate detailed results here. This paragraph should be much more condensed. Just present briefly significant observations.

Thank you for point this out. We re-wrote this paragraph.
Corrected text: Choosing the exact moment to submit a patient with COVID-19 in respiratory distress to endotracheal intubation and mechanical ventilation is still a challenge to emergency physicians and intensivists. In this cohort study, we demonstrate that time of intubation affects outcomes, and the adjusted hazard ratio for mortality was 1.018 for each day of delay in intubation following the onset of symptoms.

Page 5, line 237-238: As I realize your study concerns not typical ED or ICU but rather covid-19 unit. Please delete this sentence as it is not necessary and informative at all.

Thank you for point this out. You are completelly right! We deleted the sentence.

Page 5, line 242-244: Standardization of therapy does not mean that every patient receives the same medication or is mechanically ventilated in the exactly the same way (parameters). Usually the therapeutic strategy is individualized to some extent. Please change the meaning of this paragraph as it is incorrect.

Thank you for point this out. We rewrote the paragraph.
Corrected text: On the other hand, this study has some very important strengths. The number of patients enrolled is higher than most of the studies approaching this issue. The fact that it was performed in a single center is more a strength than a weakness. Our hospital was dedicated to assist only severe COVID-19 patients. At the pandemics peak, we had 350 ICU beds for these patients. The track record of our ICUs were comparable to other centers around the world (28), confirming that the ICU care was not determinant for the outcome. 

Reviewer 2 Report

The introduction, methods and results are clearly described.  I just want to make two suggestions.

In the methods, the authors do not include variables in their study that have shown, in some articles, association or tendency to poor prognosis as ROX index, SOFA score or obesity. Can they explain why have not been included in the analysis?

The discussion can be improved because the authors do not compare their results with recent data published, most of them showing that timing of intubation for patients with severe COVID-19 pneumonia was not significantly associated with overall mortality. The articles below show these findings and probably may be incorporated to this original.

Effect of Intubation Timing on the Outcome of Patients With Severe Respiratory Distress Secondary to COVID-19 Pneumonia. Fayed M, Patel N, Yeldo N, Nowak K, Penning DH, Vasconcelos Torres F, Natour AK, Chhina A. Cureus. 2021 Nov 16;13(11):e19620. doi: 10.7759/cureus.19620

Invasive Airway "Intubation" in COVID-19 Patients; Statistics, Causes, and Recommendations: A Review Article.Mohammadi M, Khafaee Pour Khamseh A, Varpaei HA. Anesth Pain Med. 2021 Jul 9;11(3): e115868. doi: 10.5812

Author Response

Dear Revisor,

Thank you for your suggestions, comments and references for manuscript improvements!
Below we provide a detailed point-by-point response to you. Original comments will be provided and our response will be immediately below.

1. The introduction, methods and results are clearly described.  I just want to make two suggestions. In the methods, the authors do not include variables in their study that have shown, in some articles, association or tendency to poor prognosis as ROX index, SOFA score or obesity. Can they explain why have not been included in the analysis?

Thank you for point this out! Unfortunalately, we do not have some suggested data, for example: weight, height and we also do not have all SOFA laboratory tests of all patients.

The discussion can be improved because the authors do not compare their results with recent data published, most of them showing that timing of intubation for patients with severe COVID-19 pneumonia was not significantly associated with overall mortality. The articles below show these findings and probably may be incorporated to this original.

Effect of Intubation Timing on the Outcome of Patients With Severe Respiratory Distress Secondary to COVID-19 Pneumonia. Fayed M, Patel N, Yeldo N, Nowak K, Penning DH, Vasconcelos Torres F, Natour AK, Chhina A. Cureus. 2021 Nov 16;13(11):e19620. doi: 10.7759/cureus.19620

Invasive Airway "Intubation" in COVID-19 Patients; Statistics, Causes, and Recommendations: A Review Article.Mohammadi M, Khafaee Pour Khamseh A, Varpaei HA. Anesth Pain Med. 2021 Jul 9;11(3): e115868. doi: 10.5812

Thank you for point this out! We added two studies to Introduction and Discution.
Corrected text: Fayed et al. in a single center retrospective study with 110 patients concluded that the timing of intubation for patients with severe COVID-19 pneumonia was not significantly associated with overall mortality. However, post hoc analysis of patients who had SOFA scores between 0 and 9 showed that significantly more patients within this score range who were intubated late died compared to those who were intubated early. Additionally, Kaplan-Meier survival analysis stratified by SOFA score also showed that patients who were intubated later were more likely to die. (23)